# Harnessing hybrid perception on multi-scale features for hand-foot-mouth disease multi-region prediction based on Seq2Seq

**Bingbing Lei**[1,2]*, **Xuanjun Zhu**[1], **Tao Zhou**[1,2], **Yuxi Zhang**[1]

**1** School of Computer Science and Engineering, North Minzu University, Yinchuan, Ningxia, China, **2** Key Laboratory of Image and Graphics Intelligent Processing of State Ethnic Affairs Commission, North Minzu University, Yinchuan, Ningxia, China

* x_generation@126.com

**Data availability statement:** The data underlying the experiment in the study are available from the National Institute of Infectious Diseases of Japan

## Abstract

Accurate prediction of Hand, Foot, and Mouth Disease (HFMD) is crucial for effective epidemic prevention and control. Existing prediction models often overlook the cross-regional transmission dynamics of HFMD, limiting their applicability to single regions. Furthermore, their ability to perceive spatio-temporal features holistically remains limited, hindering the precise modeling of epidemic trends. To address these limitations, a novel HFMD prediction model named Seq2Seq-HMF is proposed, which is based on the Sequence-to-Sequence(Seq2Seq) framework. This model leverages hybrid perception of multi-scale features. First, the model utilizes graph structure modeling for multi-regional epidemic-related features. Secondly, a novel Spatio-Temporal Parallel Encoding(STPE) Cell is designed; multiple STPE Cells constitute an encoder capable of hybrid perception across multi-scale spatio-temporal features. Within this encoder, graph-based feature representation and iterative convolution operations enable the capture of cumulative influence of neighboring regions across temporal and spatial dimensions, facilitating efficient extraction of spatio-temporal dependencies between multiple regions. Finally, the decoder incorporates a frequency-enhanced channel attention mechanism(FECAM) to improve the model's comprehension of temporal correlations and periodic features, further refining prediction accuracy and multi-step forecasting capabilities. Experimental results, utilizing multi-regional data from Japan to predict HFMD cases one to four weeks ahead, demonstrate that our proposed Seq2Seq-HMF model outperforms baseline models. Additionally, the model performs well on single-region data from a city in southern China, confirming its strong generalization ability.

## Introduction

Hand-foot-mouth disease (HFMD) is an infectious condition primarily caused by various enteroviruses, primarily occurring in children under the age of 5 [1–4]. It has a rapid onset and is prone to complications, which could be life-threatening in severe cases. The incidence

(https://id-info.jihs.go.jp/surveillance/idwr), the Japan Meteorological Agency (https://www.data.jma.go.jp/risk/obsdl/index.php) and the China NCDC (https://doi.org/10.12213/11.A0006.202306.2.V1.0). The organized data can be found at https://github.com/zxj550702/hfmd_predicate/tree/master/data.

**Funding:** Bingbing Lei, Scientific research project of Ningxia Education Department (NYG2024084): This funder supported the study and contributed to the study design through providing ideas for the research framework. Tao Zhou, National Natural Science Foundation of China (62062003): This funder supported the study and contributed to the data analysis and figure preparation by providing guidance.

**Competing interests:** The authors declare that they have no competing interests.

of HFMD has been notably high in East Asia in recent years, imposing a significant economic burden and representing a major threat to public health. This is exemplified by China, which has reported over one million cases annually for the past decade [5]. Furthermore, HFMD surveillance data from Japan's National Institute of Infectious Diseases (NIID) indicate that the total reported cases in Japan over the past decade have also reached approximately two million. Therefore, HFMD has become one of the significant public health threats in East Asia. At present, no effective vaccine or specific therapy exists for HFMD [6]. Consequently, the development of accurate and reliable HFMD epidemic prediction models is of paramount importance. Such accurate forecasts enable public health authorities to make informed decisions, allowing them to implement timely and effective prevention and control strategies in line with the epidemic's trajectory. This serves to protect public health and safety and mitigate economic consequences.

The prediction of HFMD case numbers involves collecting historical data on HFMD cases in each region, along with factors that influence the disease's incidence. This process includes analyzing and evaluating these factors to ultimately forecast future occurrences of HFMD. Existing prediction methods could be divided into two categories: statistical methods and machine learning methods.

The main statistical method for predicting the number of HFMD cases is to use the Autoregressive Integral Moving Average model(ARIMA) and the Seasonal Autoregressive Integral Moving Average model (SARIMA) to capture the seasonal trend in the time series data [7,8]. At the same time, various linear regression models have also attracted the attention of researchers, such as the logarithmic regression model [9]. These models have positive implications for capturing seasonal trends in time series.

However, the spread of HFMD is influenced by many factors. Recent studies have found that the spread of HFMD is related to weather factors such as temperature, humidity, and rainfall measurement. These meteorological variables have positive or negative cumulative effects on the incidence of HFMD to varying degrees [10–13]. In addition, air pollution factors such as PM10, SO2, and NO2 have also been shown to have an impact on the transmission of HFMD [14–16]. The above factors should be considered comprehensively in the task of predicting HFMD. As a linear model, statistical methods are limited in their ability to capture nonlinear relationships. Machine learning methods provide a more robust framework for identifying complex interdependencies and latent patterns among features. Machine learning models such as Random Forest Regression(RFR) and Extreme Gradient Boosting(XGBoost) were applied to predict the monthly number of HFMD cases [17]. Also, the additive model, RFR model, and Support Vector Regression(SVR) model were compared and analyzed for their performance in predicting the daily incidence of HFMD [18].

As deep learning advances, various models capable of capturing intricate relationships between features have yielded impressive results in HFMD prediction. Long short-term memory model [19], DA-RNN model [20] and Seq2Seq-attention model [21] were used to predict HFMD cases. At the same time, some researchers attempted to integrate the statistical model with the machine learning model to form a mixed model [22–24], in order to improve the prediction accuracy of HFMD cases by combining the advantages of the two. It is worth noting that a study employed the Spatio-Temporal Graph Convolution Network (STGCN) model [25] for predicting the incidence of HFMD cases. This approach not only utilized the time series data of the local city but also considered the influence of the epidemic in neighboring cities.

The aforementioned research has significantly advanced HFMD prediction and prevention by providing theoretical frameworks and technical tools that enable regions to implement effective prevention and control strategies informed by epidemic predictions. Nevertheless,

the spread of HFMD is a dynamic spatial process, characterized by randomness, uncertainty, and intricate spatio-temporal fluctuations. Traditional prediction methods often struggle to accurately capture the characteristics of multi-regional HFMD epidemic transmission simultaneously. And these methods often overlook the extraction of frequency domain features. Frequency domain features can unveil the periodicity, seasonality, and relative intensity of various frequency components, which are challenging to directly observe in the time domain. These features offer crucial information that enhances the prediction accuracy of HFMD models. Therefore, it is crucial to develop a methodology that effectively captures nonlinear associations and spatio-temporal dependencies to improve prediction accuracy. Specifically, the core challenge is developing a model that effectively integrates multiple factors, enabling it to capture spatio-temporal dependencies, perform spatio-temporal inference, and account for key influencing variables concurrently. Moreover, simultaneous prediction of epidemic trends across multiple regions, along with multi-step prediction(e.g. , short-, medium-, and long-term), would better support the dynamic allocation of prevention and control resources and the development of emergency plans.

To this end, this paper proposes the Seq2Seq-HMF model, which harnesses hybrid perception of multi-scale features for multi-region HFMD prediction. By considering the three dimensions of time, space, and frequency with multiple scale, historical HFMD cases and meteorological conditions are utilized as features to predict future HFMD cases. The main contributions of this paper are summarized as follows:

(1) The Seq2Seq-HMF model is proposed for the multi-region, multi-step prediction of HFMD cases. It features an encoder equipped with spatial-temporal parallel encoding(STPE) cells. This module functions as an encoder to simultaneously extract the time series data of the target city and incorporate the cumulative spatial impact of neighboring cities, thereby facilitating a more comprehensive prediction of HFMD.

(2) A frequency-enhanced Channel Attention mechanism (FECAM) for HFMD prediction is introduced [26]. The FECAM module models the frequency correlation between channels based on discrete cosine transform, which improves the ability of the model to extract frequency features, thus enhancing the accuracy of multi-step prediction.

(3) Comparative experiments using real-world data collection demonstrate that Seq2Seq-HMF performs better in multi-region, multi-step prediction tasks. Ablation studies further validate the effectiveness of each module.

## Related work

### Time series prediction

Time series prediction models have a long and rich research history, with applications spanning various domains such as finance, meteorology, and healthcare. Traditional statistical models, such as ARIMA [27] and SARIMA [28], have been widely used for time series prediction tasks, including the prediction of HFMD incidence. These models primarily focus on modeling the temporal dependencies within a single time series, rather than explicitly capturing complex relationships between multiple, potentially related, time series. To achieve better predictive performance, machine learning methods such as SVR, RFR [29], and XGBoost [30] have been employed to model the nonlinear correlations within the data.

Over the past decade, deep learning methods have been increasingly adopted for HFMD prediction, with RNN-based models being prominent examples. Typically, RNN-based methods employ a recurrent architecture to model the transition of temporal states [31]. However, traditional RNN suffer from issues such as gradient vanishing and gradient explosion, which limit their effectiveness in long-term prediction [32]. To address these issues, variants

of RNNs, such as the Long Short-Term Memory (LSTM) model [33] and the Gated Recurrent Unit (GRU) model [34], have been developed. These models utilize memory and forgetting mechanisms to decide whether to retain or discard information, thereby mitigating the problems associated with traditional RNNs. The Bidirectional LSTM (BiLSTM) is an extension of LSTM that processes the input sequence in both forward and backward directions. This allows it to capture dependencies from both past and future context, effectively extracting features from both directions, and achieves better performance than LSTM in general time series prediction tasks [35].

Furthermore, the Seq2Seq [36] model, which consists of an encoder-decoder architecture, has been applied to time series prediction tasks. The encoder processes the input sequence and compresses the information into a context vector, which is then used by the decoder to generate the output sequence. This architecture allows for the modeling of complex dependencies and interactions within the data, facilitating the learning of long-term dependencies and generally achieving competitive performance in HFMD prediction tasks. In addition to these models, other advanced techniques such as attention mechanisms have also shown promise in time series prediction. These mechanisms allow the model to focus on different parts of the input sequence when generating the output, thereby improving the accuracy of predictions.

In summary, while traditional statistical and machine learning methods have laid a strong foundation for time series prediction, the advent of deep learning, particularly CNN-based and RNN-based models and their variants, has significantly advanced the field. The application of these models to HFMD prediction represents a crucial step in leveraging the broader advancements in time series analysis to address specific public health challenges.

## Graph convolution neural network

Graph Convolution Networks (GCN) were proposed as a solution to the expensive computational costs of GNN [37]. It extracts higher-level features of the target node by aggregating information from neighboring nodes and the node itself, thereby capturing local structural information in the graph. Meanwhile, the information of edges in the graph structure can also be added as supplementary features to the node's calculation. The new state of each node is obtained by operating the input features and the adjacency matrix, and this operation can be regarded as a local convolution on the graph [38–40]. The specific operation of GCN can be expressed by the following formula:

$$H^{n+1} = \sigma\left(\hat{D}^{-\frac{1}{2}} \hat{A} \, \hat{D}^{-\frac{1}{2}} H^n W^n\right)) \tag{1}$$

where $H^n$ represents the n-TH layer feature matrix, which gathers the information of n-hop adjacent nodes. Stands for adjacency matrix. Represents the matrix with self-loops, I represents the identity matrix, $\hat{D}$ is the degree matrix of $\hat{A}$, and $W^n$ represents the weight matrix of the n-TH layer, $\sigma$ is the Sigmoid activation function. In particular, when $n = 0$, represents the input data $X$. The list of symbols used throughout this paper is provided in Table 1.

## Materials and methods

### Definition of problem

In this paper, the city node graph is conceptualized as G = (V, E), where V consists of N city nodes and E is the set of connections between nodes. For each node, The dynamic HFMD multi-feature set, $X = \{x_1, x_2, ..., x_t\}$, represents the state of the multi-feature set across t time

**Table 1. A summary of symbols and descriptions.**

| Symbol | Description |
|--------|-------------|
| $G$ | A graph structure composed of all nodes |
| $V$ | Node set |
| $E$ | Edge set |
| $N$ | Number of nodes |
| $F$ | The number of features per node |
| $X$ | Initial input for each module |
| $S$ | The length of the entire input time series |
| $L$ | The length of the prediction |
| $R$ | Representational reality feature |
| $M$ | Test set time series length |
| $W$ | The weight matrix to be learned |
| $w$ | The size of the time window |
| $b$ | Bias matrix to be learned |
| $Y$ | A matrix of predicted values |
| $h$ | The state vector representing the output in the formula |
| $y$ | The true number of HFMD cases |
| $\hat{y}$ | The predicted number of HFMD cases |
| $K$ | The number of parameters of the model |
| $\odot$ | Element-wise multiplication |

points. In this paper, the multi-variable multi-step prediction method is used. The multi-feature set includes not only the number of HFMD cases but also the meteorological conditions that affect the transmission of HFMD. The final input sequence is $X \in R^{S \times F}$, where S represents the total length of the entire input sequence and F represents the number of features. In the prediction work, a sliding window is used to continuously sample the input sequence.

Specifically, given input data $X$, it is divided into $\frac{S-w}{L}$ sub-sequences using a sliding window of size $w$ and considering a historical length of $t$. For each sub-sequence $X^{(t-w+1:t)}$, the number of HFMD cases over the next $L$ time points is taken as the ground truth. The model is expected to learn a parameterized mapping function $f$ during the training process to perform the multi-region multi-step prediction task of HFMD:

$$Y_{t+1}, \ Y_{t+2}, \ \cdots, \ Y_{t+L} = f\big(G; (x_{t-n}, \ \dots, \ x_{t-1}, \ x_t)\big) \tag{2}$$

## Model architecture

Accurate prediction the number of HFMD cases necessitates comprehensive analysis of both temporal dynamics and spatial interdependencies, particularly in modern urban networks with intensive population mobility. While existing prediction models predominantly focus on single-location temporal patterns, they often neglect the critical spatial correlations arising from inter-city connectivity. This study proposes Seq2Seq-HMF, a novel Seq2Seq-based framework that addresses the spatiotemporal heterogeneity in multi-regional HFMD transmission through hybrid perception mechanisms. Model integrates multi-scale weather features with urban interaction patterns to model the complex epidemiological relationships across geographical nodes, thereby overcoming the spatial-temporal fragmentation limitations in current prediction methodologies.

The overall architecture of the Seq2Seq-HMF model is shown in Fig 1. It uses the Seq2Seq model as the baseline, which is based on an encoder-decoder architecture to convert the input sequence into the output sequence. The Seq2Seq-HMF consists of the following two parts: 1) an encoder with STPE cells; and 2)a decoder containing FECAM.

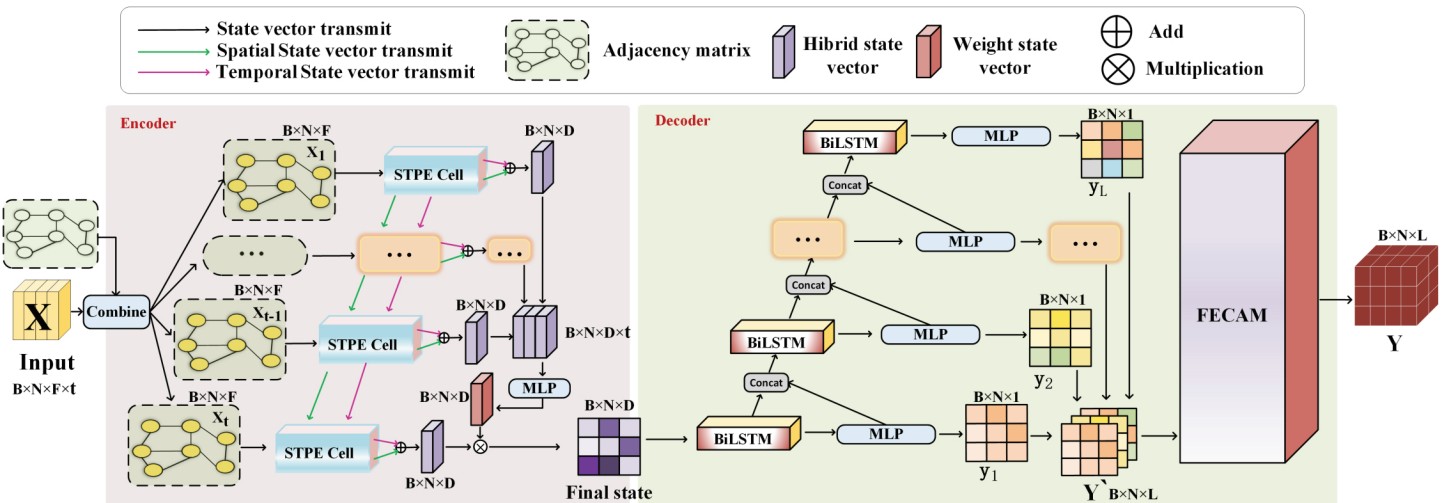

**Fig 1. The overall structure of Seq2Seq-HMF.** It comprises an encoder and a decoder. The encoder performs multi-scale hybrid perception on input time series data from multiple regions to extract features, which the decoder then processes to generate predictive time series data for those regions.

Firstly, the input data consists of two parts. The first part is the input features: the multidimensional data is classified by city nodes, and each node contains its multidimensional data features; The second is the adjacency matrix of the graph, which together with the input features of all nodes forms a complete graph structure. The graph structure of $t$ time makes up the final input. That is, the $X$ shape of the input model is $B \times N \times F \times t$, where $B$ is the batch size, $N$ is the number of nodes, $F$ is the number of features, and $t$ is the point of time.

Subsequently, the data corresponding to each time is passed into the corresponding STPE Cell to extract the spatial and temporal correlation between the data. Secondly, two state vectors output by each STPE Cell are connected by residuals to obtain the space-time-dependent state vector. To obtain a more fine-grained, multi-scale representation of space-time features, the state vectors corresponding to the first to $t - 1$ time are stacked and passed into the multi-layer perception(MLP) layer to learn a weight state vector. After multiplying the result with the state vector corresponding to the $t$ time, the Final state vector containing multi-scale information is obtained.

The Final state is then passed into the decoder module. After each decoder has finished decoding, the state vector is transformed by a multi-layer MLP into a predicted value $y$ that encompasses all nodes. Next, the predicted value $y$ is concatenated with the updated state vector from the current decoder and then passed to the subsequent decoder. When the predicted values of $L$ moments are obtained, they are stacked to form $Y'$. Finally, $Y'$ is processed through the FECAM module for frequency enhancement, yielding the final predicted value $Y$. Here, $Y$ represents the number of HFMD cases over a continuous span of $L$ future time points, encompassing N nodes.

## Encoder with spatial-temporal parallel encoding cells

As shown in Fig 2, each STPE Cell contains a temporal graph convolutional network [41](TGCN) and a bidirectional long short-term memory network (BiLSTM).

The TGCN (Fig 2(A)) module is used to encode spatial dimensions, which can simultaneously capture dynamic transformations of topological spatial correlation and time series data to obtain persistent shadows between different nodes. BiLSTM (Fig 2(B)) encodes from

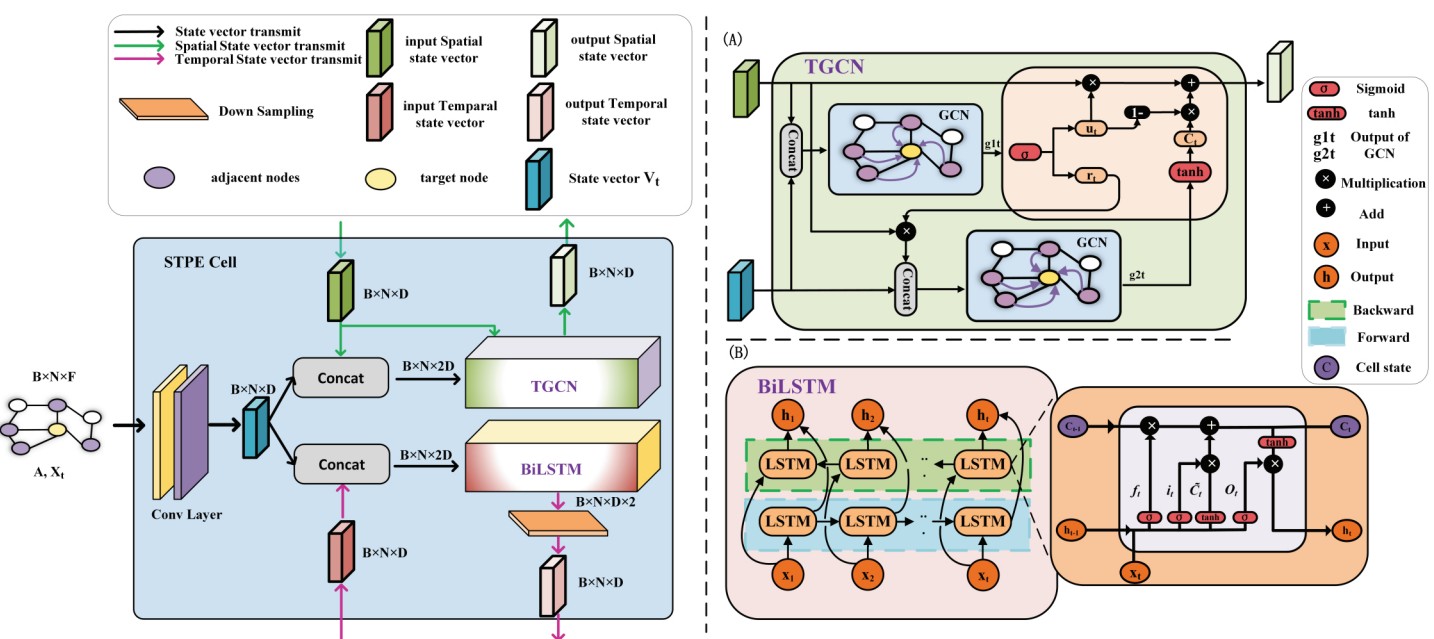

**Fig 2. Structure of STPE Cell.** It is contains TGCN (A) and BiLSTM (B). Both modules receive the output of the previous STPE Cell in addition to the data features.

the time dimension, captures the time correlation between each node's data, enriches the time autocorrelation of nodes, and operates in parallel.

**Spatial encoding.** The operation of the SPTE Cell proceeds as follows: Given the input $X \in R^{S \times F}$ and the adjacency matrix A of its nodes. The encoder has w STPE cells, and each STPE cell has an independent state parameter. Assume that at time $t$, $X_t$ is the multi-feature set of all current nodes, and $h_{t-1}$ is the input state vector encoded at the previous time. $X_t$ is input into a one-dimensional Convolution Layer, and the feature channels of each node are convolved separately to map to a higher-dimensional space to generate a state vector $V_t$ of size D. If the previous STPE Cell outputs Spatial state vector and Temporal state vector, $V_t$ is spliced with them respectively to form a new state vector $V_t$:

$$V_t = \begin{cases} \text{Conv1D}(X_t), & i = 1 \\ \text{Concat}(h_{t-1}, \text{Conv1D}(X_t)), & 1 < i < w \end{cases} \tag{3}$$

Then enter the state vector $V_t$ into TGCN and BiLSTM modules respectively. Let $f(X_t, A)$ represent the process of graph convolution as shown in Eq (1), TGCN state transformation equation is as follows:

$$\begin{aligned} g_{1_t} &= f([V_t, h_{t-1}], A) \\ u_t &= \sigma\left(W_t^u [g_{1_t}, V_t] + b_u\right) \\ r_t &= \sigma\left(W_t^r [g_{1_t}, V_t] + b_r\right) \\ g_{2_t} &= f([V_t, (r_t \odot h_{t-1})], A) \\ c_t &= \tanh\left(W_t^c g_{2_t} + b_c\right) \\ h_t &= u_t \odot h_{t-1} + (1 - u_t) \odot c_t \end{aligned} \tag{4}$$

Where $g1_t$, $g2_t$ represent the output result of the graph convolution at time t. $u_t$ is the update gate, $r_t$ is the reset gate, is the candidate hidden state vector, $h_t$ is the output Spatial state vector at time $t$, and tanh is the activation function. For each GCN operation, various features of the target node and neighbor node are aggregated to update the incoming state vector. With the goal of capturing the persistent influence from neighboring nodes, after extracting spatial correlation information through graph convolution operations, the state vector g1t and g2t is fed into the GRU to model temporal dependencies.

**Temporal encoding.** While TGCN obtains the long-term dependence of the target node on the neighbor node, the time autocorrelation of the target node is obtained by the BiLSTM module. Based on the LSTM model, further strengthens the ability to capture the information before and after the sequence data by introducing the forward and backward mechanism.

Specifically, the operation process of the LSTM module at time $t$ can be expressed as:

$$
\begin{aligned}
i_t &= \sigma\left(W_t^i\left[h_{t-1}, X_t\right] + b_i\right) \\
f_t &= \sigma\left(W_t^f\left[h_{t-1}, X_t\right] + b_f\right) \\
\widetilde{c}_t &= \tanh\left(W_t^c\left[h_{t-1}, X_t\right] + b_c\right) \\
o_t &= \sigma\left(W_t^o\left[h_{t-1}, X_t\right] + b_o\right) \\
c_t &= f_t \odot c_{t-1} + i_t \odot \widetilde{c}_t \\
h_t &= o_t \odot \tanh(c_t)
\end{aligned}
\tag{5}
$$

Where $i_t$ is the input gate, $f_t$ is the forgetting gate, and $o_t$ is the output gate. BiLSTM consists of bidirectional LSTM layers, as shown in Fig 2(B), $X_1$, $X_2$, ... $X_t$ represents the corresponding input data at each moment, and is passed into two LSTM layers, the hidden state will be merged into $h_0$, $h_2$, …, $h_t$ as the corresponding output data.

Let LSTM($h_{t-1}$, $X_t$) represent the operation process of equality group (5), then the operation process of BiLSTM can be expressed as:

$$
h_t = Concat(LSTM(h_{t-1, forward}, X_t), LSTM(h_{t+1, backward}, X_t))
\tag{6}
$$

Where $h_t$, $h_{t-1}$, and $h_{t+1}$ indicate the output Temporal state vector at the corresponding time respectively, $X_t$ indicates the input state vector at time $t$, and forward and backward indicate the transmission direction of the state vector.

## Decoder containing FECAM

**Time series decoder.** Seq2Seq-HMF uses BiLSTM as the decoder to decode the Final state. The number of decoders corresponds one-to-one with the predicted length. Let BiLSTM(h) represent decoding the input state vector according to formula group (6). Then the decoding process at a certain prediction time can be expressed as:

$$
\begin{aligned}
h_{l+1} &= \mathrm{BiLSTM}([h_l, y_l]) \\
y_{l+1} &= \mathrm{ReLU}(\mathrm{Dropout}(W_{l+1} h_{l+1}))
\end{aligned}
\tag{7}
$$

Where $h_l$ represents the state vector obtained by decoder decoding, and $y_l$ represents the frequency enhancement vector at a certain predicted time. The composition of the MLP is as follows: $W_{l+1}$ represents the weight matrix of the fully connected layer, Dropout represents the neurons that have lost a certain percentage, and RuLu is the activation function. In particular, when $l = 0$ indicates the Final output state of the encoder.

**Frequency enhanced channel attention mechanism.** The structure of FECAM is shown in Fig 3. Firstly, the input $Y'$ is divided into a one-dimensional vector v according to the node dimension, and then the frequency features are extracted by DCT operation for each one-dimensional vector in turn. After that, the frequency vectors are stacked by node dimension to form the entire frequency tensor Freq, and the MLP is passed to learn the frequency dependence between different nodes. Finally, the resulting attention matrix $F_{att}$ is multiplied element by element with the original eigenvector to obtain the frequency-enhanced predicted value $Y$.

Among them, the process of DCT transformation of one-dimensional vector v is as follows:

$$f_k^{1d} = \alpha_k \sum_{i=0}^{L-1} v_i^{1d} \cos\left(\frac{\pi}{L}\left(i + \frac{1}{2}\right)k\right), \quad \alpha_k = \begin{cases} \sqrt{\frac{1}{L}} & \text{if } k = 0 \\ \sqrt{\frac{2}{L}} & \text{if } k > 0 \end{cases} \tag{8}$$

$$v^{DCT} = \text{stack}(f_k^{1d})$$

Where $v_i^{1d}$ is the I-th element of the original sequence v, $f_k^{1d}$ is the spectral coefficient of the K-th element after the transformation, L is the length of the sequence, $i, k = 0, 1, ..., L - 1$. $\alpha_k$ is the normalization factor, ensuring that the transformation is orthogonal, cos is the cosine function, and $v^{DCT}$ represents the result of the transformation of the one-dimensional sequence $v$.

In this paper, the spatio-temporal dependent state vector obtained by the decoder is divided into N channels according to city nodes to get the channel vector $v_i, i \in (1, N)$. Let DCT(v) represent the transformation process of equation group (8), then the overall operation flow of FECAM is as follows:

$$v_1, v_2, ..., v_N = \text{Split}(Y'_{1:N})$$
$$v_i^{DCT} = \text{DCT}(v_i)$$
$$Freq = \text{stack}(v_i^{DCT}) \tag{9}$$
$$F_{att} = \sigma(W_2 \delta W_1 Freq)$$
$$Y = Y' \odot F_{att}$$

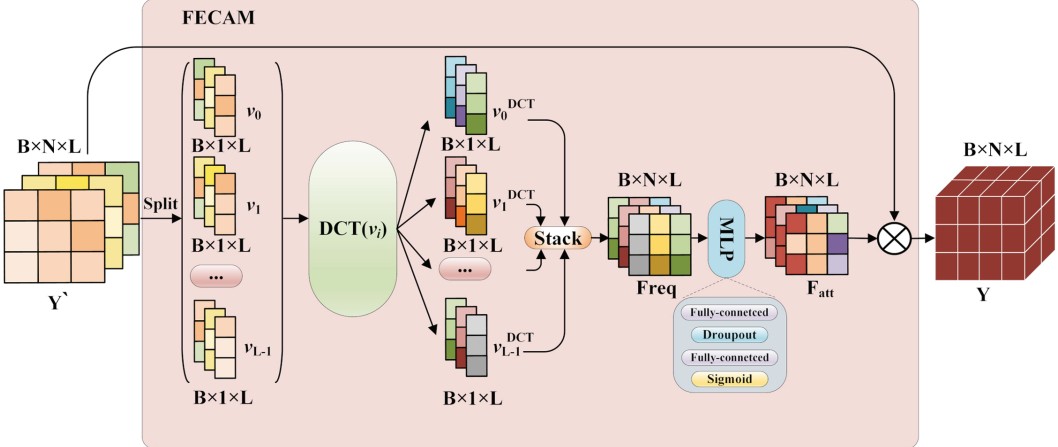

**Fig 3. structure of FECAM.** It is consists of operations such as DCT, stack, MLP, and multiplication.

After each frequency channel vector is obtained, it is stacked to obtain the tensor Freq, and then the frequency dependence between different channels is established using the full connection layer learning channel attention $F_{att}$. Ultimately, the input vector is element-wise multiplied by the channel attention weights, yielding a weighted representation that yields the final predicted value $Y$. This process ensures that the network layer's output aligns with the frequency domain characteristics of the input data.

The above operations finely extract frequency domain information from each channel, allowing the frequency features of each channel to interact with one another. This process yields more comprehensive frequency domain information, further enhancing the model's feature extraction and characterization capabilities, and thereby improving the accuracy of predictions.

## Loss function

To minimize the error between the real value and the predicted value of the number of HFMD cases in the training process, the loss function used in this paper is as follows:

$$Loss = \frac{1}{L} \sum_{n=1}^{N} \sum_{l=1}^{L} (y_{n,l} - \hat{y}_{n,l})^2 + \lambda \sum_{k=1}^{K} \omega_m^2 \tag{10}$$

The first term on the right-hand side of the equation is the mean squared error (MSE), and the second term is the L2 regularization term, which can effectively prevent overfitting. $y_{n,l}$ represents the actual value, $\hat{y}_{n,l}$ represents the predicted value. This paper predicts the number of HFMD cases for N urban nodes, and when calculating Loss, the loss value is computed for all nodes across all predicted time points. The $\lambda$ coefficient is used to control the strength of the regularization, $\omega$ represents the weight parameters of the model, and $K$ is the number of model parameters.

## Evaluation metrics

To fairly evaluate the performance of the model, this paper uses the Root Mean Square Error (RMSE), Mean Absolute Error (MAE), and Coefficient of Determination ($R^2$) to measure the discrepancy between the true values and the predicted values. The three metrics are defined as follows:

$$RMSE = \sqrt{\frac{1}{NM} \sum_{n=1}^{N} \sum_{m=1}^{M} (y_{n,m} - \hat{y}_{n,m})^2} \tag{11}$$

$$MAE = \frac{1}{NM} \sum_{n=1}^{N} \sum_{m=1}^{M} |y_{n,m} - \hat{y}_{n,m}| \tag{12}$$

$$R^2 = 1 - \frac{\sum_{n=1}^{N} \sum_{m=1}^{M} (y_{n,m} - \hat{y}_{n,m})^2}{\sum_{n=1}^{N} \sum_{m=1}^{M} (y_{n,m} - \bar{y})^2} \tag{13}$$

Where $\bar{y}$ represents the average of all true values, $y$ represents the actual value, $\hat{y}$ represents the predicted value, $N$ is the number of nodes, and $M$ is the length of the test set time series.

## Experimental results

### Data collection

The HFMD data for the multi-region prediction were obtained from the NIID of Japan, and the corresponding meteorological data is provided by the Japan Meteorological Agency, encompassing temperature, relative humidity, atmospheric pressure, wind speed, and rainfall data. This multi-region dataset consists of weekly figures from December 2013 to December 2023 for Japan's 47 prefectures. Additionally, to evaluate the model from different perspectives, a single-region dataset was utilized. This dataset includes daily HFMD cases and meteorological variables for a southern China city from January 2014 to December 2019. It was publicly released by the China National Center for Disease Control and Prevention [42]. To capture the influence of the pronounced seasonality of HFMD, the week number will be incorporated as a temporal variable. The data are divided into training and test sets at an 8:2 ratio. A a linear interpolation method is applied to the missing values. Min-Max Normalization is applied to normalize the data before it is fed into the model. The data are statistically described in Table 2.

### Experimental setup

The experimental environment consists of a Windows 11 system, with a computer memory of 16GB, and is equipped with an NVIDIA GeForce RTX 4060 Laptop GPU. The model was built using the PyTorch (GPU) framework with Python version 3.11.5, PyTorch version 2.3.1, and CUDA version 12.1. During the training process, the Adam optimizer is employed with an initial learning rate of 0. 001. The training consisted of 500 epochs. Given the model architecture wherein each layer consists of D=128 neurons, it is imperative to adjust the state vector accordingly to ensure compatibility with the specified neural network structure. The length of historical data, $t$, is set to 4, and the prediction length, $L$, varies from 1 to 4 time steps. In the comparison experiment section below, to fairly compare the performance of different models, this paper uses the model's corresponding prediction paper settings.

Comparative experiments are conducted respectively on multi-region and single-region datasets. In the multi-region dataset, 47 counties were ranked based on official data from Japan. The goal of these experiments was to compare the prediction effects of the RFR, XGBoost, TGCN, STGCN, LSTM, Seq2Seq-Shil, Seq2Seq-HMF, and DA-RNN models and to verify the validity of the Seq2Seq-HMF model. Otherwise, to clarify the impact of each component on Seq2Seq-HMF performance, the ablation study section systematically evaluates the

**Table 2. Statistical description of variables.**

| Var | Dataset | Minimum | P25 | Median | P75 | Maximum | Mean |
|---|---|---|---|---|---|---|---|
| HFMD cases | Multi-region | 0 | 5 | 20 | 67 | 4833 | 82.7 |
| Temperature (°C) | | -7.7 | 8.8 | 16.5 | 23.2 | 32.2 | 16.1 |
| Relative humidity (%) | | 31 | 64 | 71 | 77 | 97 | 70.2 |
| Pressure (hPa) | | 953.6 | 1004.8 | 1010 | 1015 | 1027.1 | 1008.4 |
| Wind speed (m/s) | | 0.8 | 2.2 | 2.7 | 3.3 | 14 | 2.9 |
| Precipitation (mm) | | 0 | 4.5 | 18.5 | 43 | 876 | 33.3 |
| HFMD cases | Singel region | 1 | 32 | 60 | 110 | 473 | 85.8 |
| Temperature | | -4.5 | 10.3 | 18.4 | 24.3 | 32.9 | 17.6 |
| Relative humidity | | 23 | 72 | 81 | 88 | 100 | 79.6 |
| Pressure | | 983 | 1008.4 | 1015.9 | 1023 | 1039.7 | 1015.8 |
| Wind speed | | 0.1 | 1.3 | 1.8 | 2.4 | 12 | 1.97 |
| Precipitation | | 0 | 0 | 0 | 3.4 | 276.2 | 4.9 |

key components of the model through ablation experiments and analyzes the experimental results on multi-region dataset.

## Comparison results and analysis

From Table 3 and Fig 4, it is observed that as the prediction step $L$ increases, the performance of all models in predicting the number of HFMD cases in both datasets gradually deteriorates across the three metrics. Compared to other models, the Seq2Seq-HMF model's performance decline is slower, enabling it to provide more accurate predictions over an extended period. Besides, TGCN and STGCN network models are lower than other models in short - and long-term predictions in a large range and multiple regions. This is because predicting infectious diseases differs from other tasks, such as traffic flow forecasting, in that it places greater emphasis on the autocorrelation of nodes. In single-region prediction tasks, GCN-based models operate solely on the target region's data. As a result, they are unaffected by the previously mentioned limitations, exhibiting adequate performance.

**Table 3. Comparison of different model predictions.**

| Model | Dataset | MAE | | | | RMSE | | | | $R^2$ | | | |
|---|---|---|---|---|---|---|---|---|---|---|---|---|---|
| | | L=1 | L=2 | L=3 | L=4 | L=1 | L=2 | L=3 | L=4 | L=1 | L=2 | L=3 | L=4 |
| RFR | Multiple | 18.56 | 23.32 | 27.69 | 35.11 | 51.62 | 63.01 | 68.49 | 88.56 | 0.796 | 0.69 | 0.646 | 0.411 |
| XGBoost | | 18.32 | 23.59 | 27.77 | 34.74 | 49.14 | 61.59 | 68.28 | 86.89 | 0.815 | 0.71 | 0.65 | 0.432 |
| TGCN | | 36.46 | 36.3 | 38.82 | 45.31 | 68.85 | 71.4 | 74.32 | 84.17 | 0.57 | 0.54 | 0.506 | 0.479 |
| STGCN | | 32.78 | 36.99 | 34.66 | 42.75 | 60.82 | 65.46 | 74.15 | 82.4 | 0.718 | 0.678 | 0.58 | 0.497 |
| LSTM | | 17.28 | 23.62 | 31.08 | 34.21 | 35.01 | 48.65 | 57.21 | 67.29 | 0.907 | 0.819 | 0.778 | 0.656 |
| Seq2Seq-Shil | | 16.58 | 21.92 | 26.03 | 28.13 | 34.13 | 44.78 | 53.92 | 61.67 | 0.911 | 0.847 | 0.778 | 0.711 |
| DA-RNN | | **14.35** | 21.19 | 25.35 | 28.37 | 33.29 | 44.42 | 53.73 | 61.87 | 0.92 | 0.851 | 0.774 | 0.712 |
| Seq2Seq-HMF | | 15.42 | **18.96** | **22.23** | **24.90** | **30.35** | **39.08** | **47.38** | **56.27** | **0.93** | **0.883** | **0.831** | **0.76** |
| RFR | Single | 9.42 | 10.05 | 10.7 | 11.17 | 13.09 | 13.78 | 14.46 | 15.18 | 0.837 | 0.82 | 0.802 | 0.782 |
| XGBoost | | 10.49 | 11.58 | 11.18 | 12.42 | 14.22 | 15.83 | 15.36 | 16.88 | 0.808 | 0.763 | 0.776 | 0.73 |
| TGCN | | 9.84 | 11.81 | 12.22 | 13.34 | 13.48 | 15.35 | 15.58 | 17.3 | 0.83 | 0.786 | 0.773 | 0.74 |
| STGCN | | 8.95 | 10.29 | 12.73 | 19.65 | 12.35 | 13.76 | 15.78 | 23.41 | 0.85 | 0.821 | 0.77 | 0.56 |
| LSTM | | 9.07 | 9.86 | 11.08 | 11.21 | 12.51 | 13.68 | 14.64 | 15.03 | 0.852 | 0.823 | 0.798 | 0.786 |
| Seq2Seq-Shil | | 9.87 | 10.02 | 10.67 | 11.67 | 13.41 | 13.68 | 14.52 | 15.0 | 0.832 | 0.82 | 0.80 | 0.788 |
| DA-RNN | | 10.54 | 18.16 | 20.1 | 21.48 | 13.66 | 20.75 | 23.08 | 24.85 | 0.83 | 0.65 | 0.58 | 0.531 |
| Seq2Seq-HMF | | **7.96** | **9.26** | **9.92** | **10.2** | **10.95** | **12.53** | **13.39** | **13.82** | **0.887** | **0.852** | **0.829** | **0.818** |

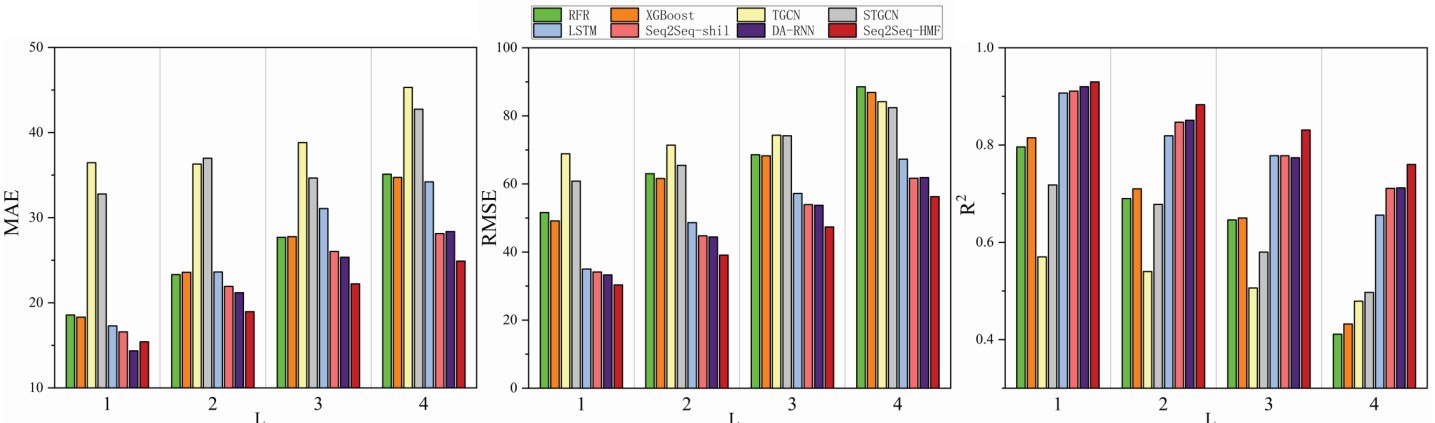

**Fig 4. Bar Chart of Performance Indicators for Model Comparison Tests in multi-region data set**.

Among the RNN-based models LSTM, Seq2Seq-shil, and DA-RNN, DA-RNN demonstrates superior performance in multi-region predictions. It employs an attention mechanism to identify the impact of weather factors on HFMD, achieving an MAE index of 14.35 for $L = 1$ predictions. Its $R^2$ is 0.92, slightly lower than Seq2Seq-HMF's 0.93. This is due to the fact that the DA-RNN model is more precise in predicting nodes with fewer cases compared to the Seq2Seq-HMF model. Conversely, the Seq2Seq-HMF model effectively captures the influence of adjacent nodes when predicting nodes with a high number of cases, resulting in more accurate predictions than DA-RNN in these instances. Overall, at the same prediction length L, the performance metrics of the Seq2Seq-HMF model surpass those of the other comparative models in both datasets. These findings indicate that the Seq2Seq-HMF model enhances the accuracy of HFMD prediction.

To directly compare model performance, this paper presents the prediction outcomes for four representative models at L=1. Fig 5 respectively shows the comparison between the predicted value and the measured value of different network models when the prediction step size $L = 1$, where the abscess Node number represents the number of city nodes, the ordinate Week represents the time series from week 1 to week 110 of the test set, and the vertical coordinate represents the value of HFMD cases. It indicates that the prediction outcomes of STGCN models is less than optimal. Specifically, during the peak period of HFMD incidence, STGCN tend to overemphasize the influence of neighboring nodes. This leads to low-incidence nodes being inappropriately affected by adjacent high-incidence nodes, resulting in a notably reduced prediction accuracy. In contrast, the DA-RNN model, all based on RNNs, have achieved higher accuracy in predicting HFMD. Particularly, the Seq2Seq-HMF model demonstrates the closest alignment between predicted results and actual values during peak incidence periods.

Moreover, several representative regions are selected, and the predicted values of HFMD for different $L$ in these regions are depicted in line charts for comparative analysis. Fig 6 and Fig 7 shows a comparison of the predicted values by each model with the actual values for a representative city in each of the eight Japanese regions: Hokkaido, Tohoku, Kanto, Chubu, Chugoku, Kinki, Shikoku, and Kyushu, based on the official regional classification. It is observed that when $L$ is greater than 1, the broken lines representing STGCN oscillate to varying degrees, and the predicted values of the XGBoost model often exhibit abnormal prediction peaks. Notably, the predicted values by STGCN tend to be higher than the ground truth, which is consistent with the aforementioned analysis. Fig 8 depicts a geospatial map representing the error ratio between the observed values and the Seq2Seq-HMF's predictions values during the HFMD epidemic peaks at weeks 43 and 96.

## Ablation Study

**Impact of spatial weight matrix.** In graph convolution operations, the choice of spatial weight matrix significantly impacts model performance. This section presents a comparative analysis of two typical spatial weight matrices—the adjacency-based spatial weight matrix and the inverse distance spatial weight matrix—focusing on their differences in model prediction accuracy.

The distance-based spatial weight matrix is employed to quantify the spatial relationships between geographical entities, among which the inverse distance spatial weight matrix is a commonly utilized approach. This matrix calculates weights based on the proximity of entities, assigning higher weights to closer entities and lower weights to more distant ones. The adjacency-based spatial weight matrix primarily considers the immediate neighborhood relationships between regions.

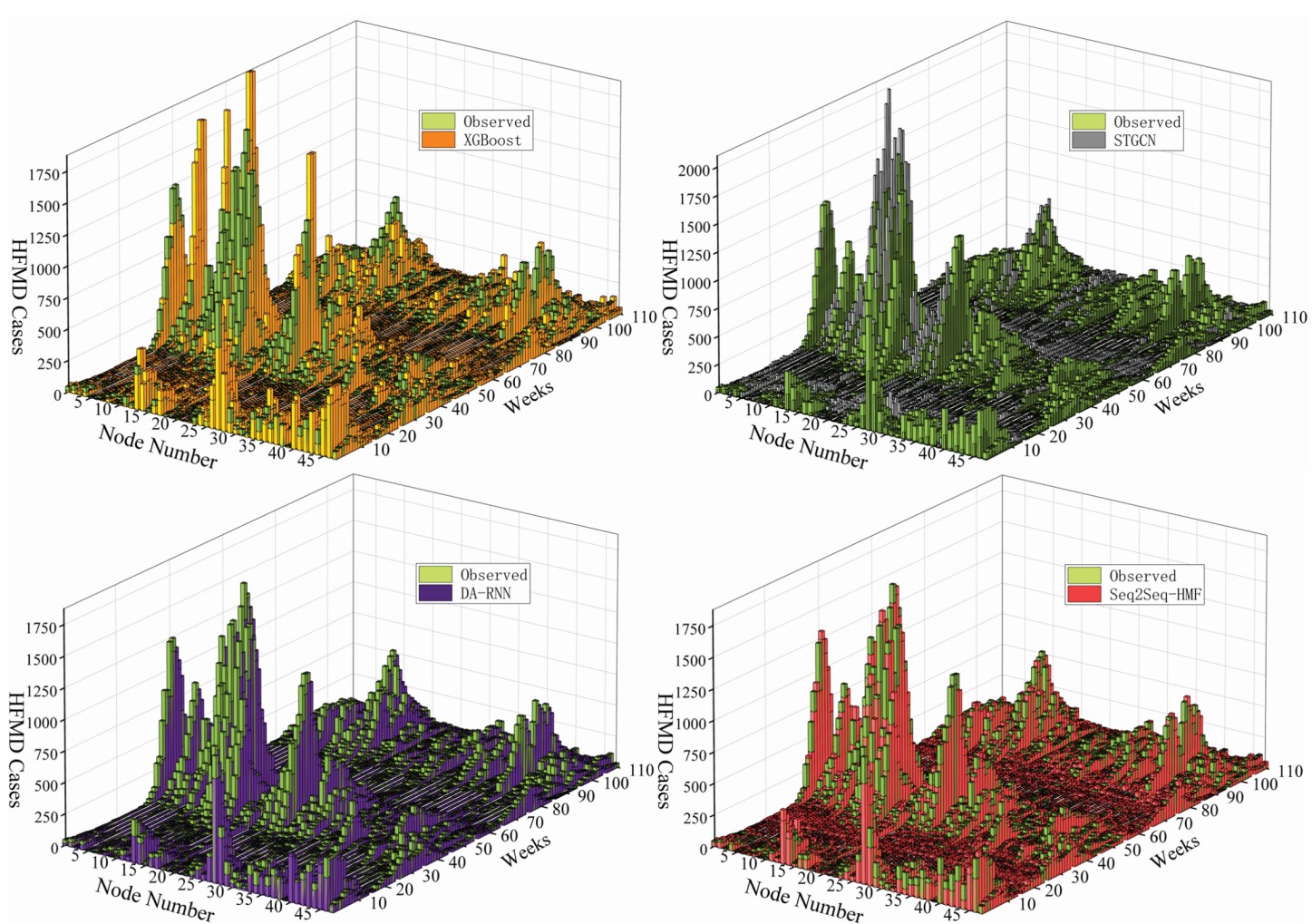

**Fig 5. 3D Spatiotemporal Distribution of HFMD Cases.** For the predicted performance of each model when L=1, the test set starts at week 46 in 2021 as the Y-axis and ends at week 51 in 2023, for a total of 110 weeks.

Table 4 shows the results indicating that the proximity-based spatial weight matrix outperforms the distance-based one in terms of prediction accuracy for HFMD propagation. This may be because disease spread prediction tasks are more sensitive to the characteristics of neighboring regions, and the distance-based weight matrix may not effectively capture this local dependency. The distance-based spatial weight matrix is less effective, possibly because it fails to fully utilize the strong correlation between neighboring regions and introduces noise from distant units. In contrast, the adjacency-based spatial weight matrix can better capture local dependencies, thus performing better in spatio-temporal feature extraction.

**Impact of sampled neighbor hop.** In the graph convolution operation, the number of hops for sampling neighbor nodes is a key hyperparameter that significantly influences the model's performance. Table 5 details the final performance of the model across various sampling hop configurations. As the number of hops for sampling neighbor nodes increases, the model's performance exhibits a nonlinear trend. The model achieves the most comprehensive optimal performance when the number of sampling hops is set to 2.

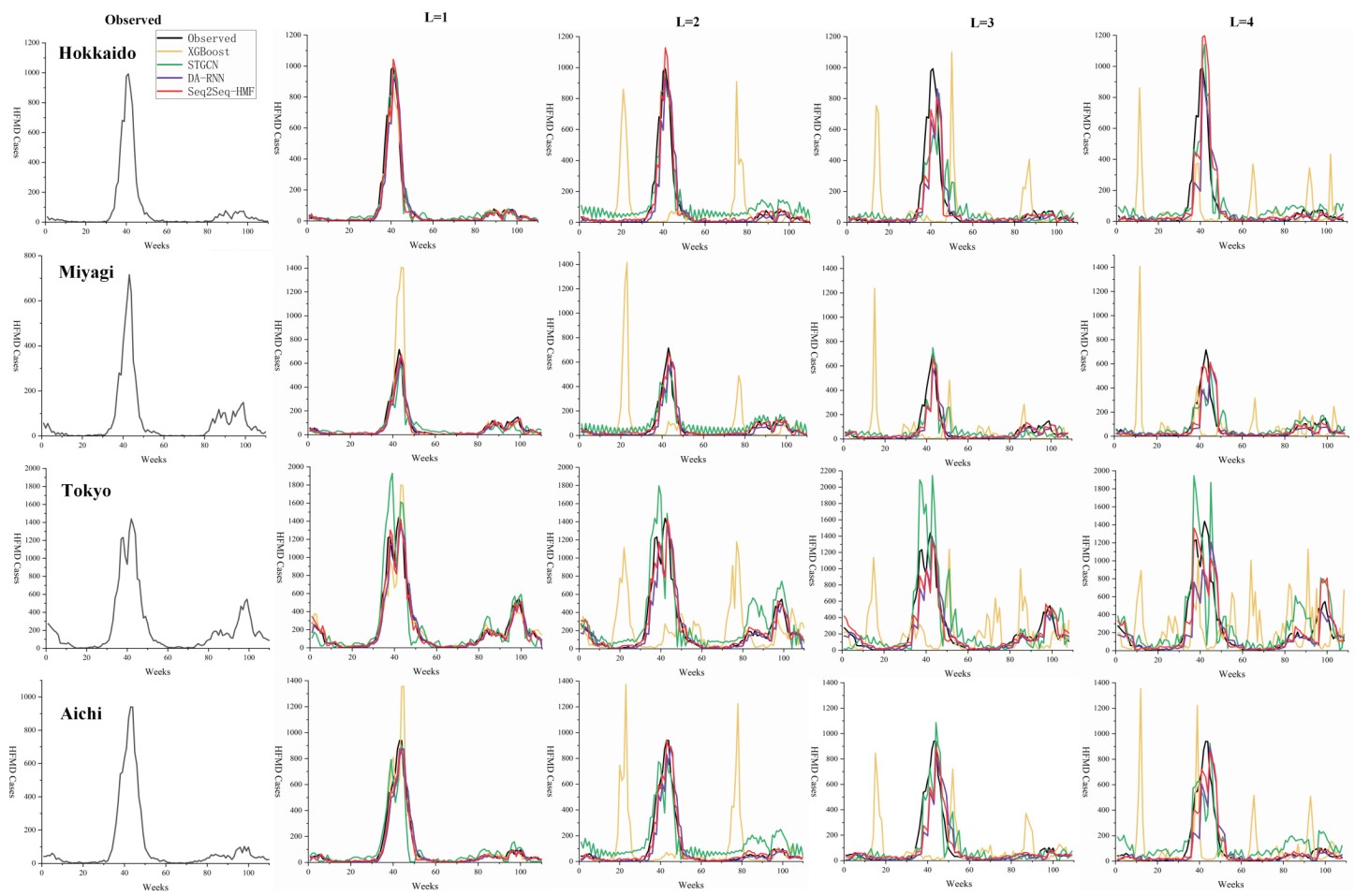

**Fig 6. Line chart of predicted values(part A).** Comparison of observed and predicted values for each representative administrative area.

**Impact of regularization parameter.** Table 6 indicates that when $\lambda$ is on the order of $10^{-4}$, setting $L$ to 1 significantly enhances the model's performance. However, when $L$ is increased to 4, the model's predictive performance deteriorates. When $\lambda$ is on the order of $10^{-3}$, the model achieves optimal performance. Furthermore, when $\lambda$ reaches the orders of $10^{-2}$ and $10^{-1}$, the model exhibits signs of underfitting.

**Impact of STPE cell and FECAM.** To validate the effectiveness and rationality of the STPE cell and FECAM modules, the following experiments have been designed in this study, as outlined in Table 7.The experimental results are shown in Table 8. EXP-1, which utilized the base model Seq2Seq, yielded slightly lower results compared to the other RNN-based models in the comparative experiment. In EXP-2, the STPE cell is employed as the encoder in the basic Seq2Seq model to extract not only the autocorrelation of features but also the persistent influence of neighboring nodes. The results indicate that all metrics are higher compared to those obtained in Experiment 1. Besides the situation at $L = 1$, significant improvements were observed in the performance of MAE, RMSE, and $R^2$. At $L = 2$, MAE decreased by 30.5% and RMSE by 8.7%, with $R^2$ increasing by 3%. At $L = 3$, MAE decreased by 28.3%

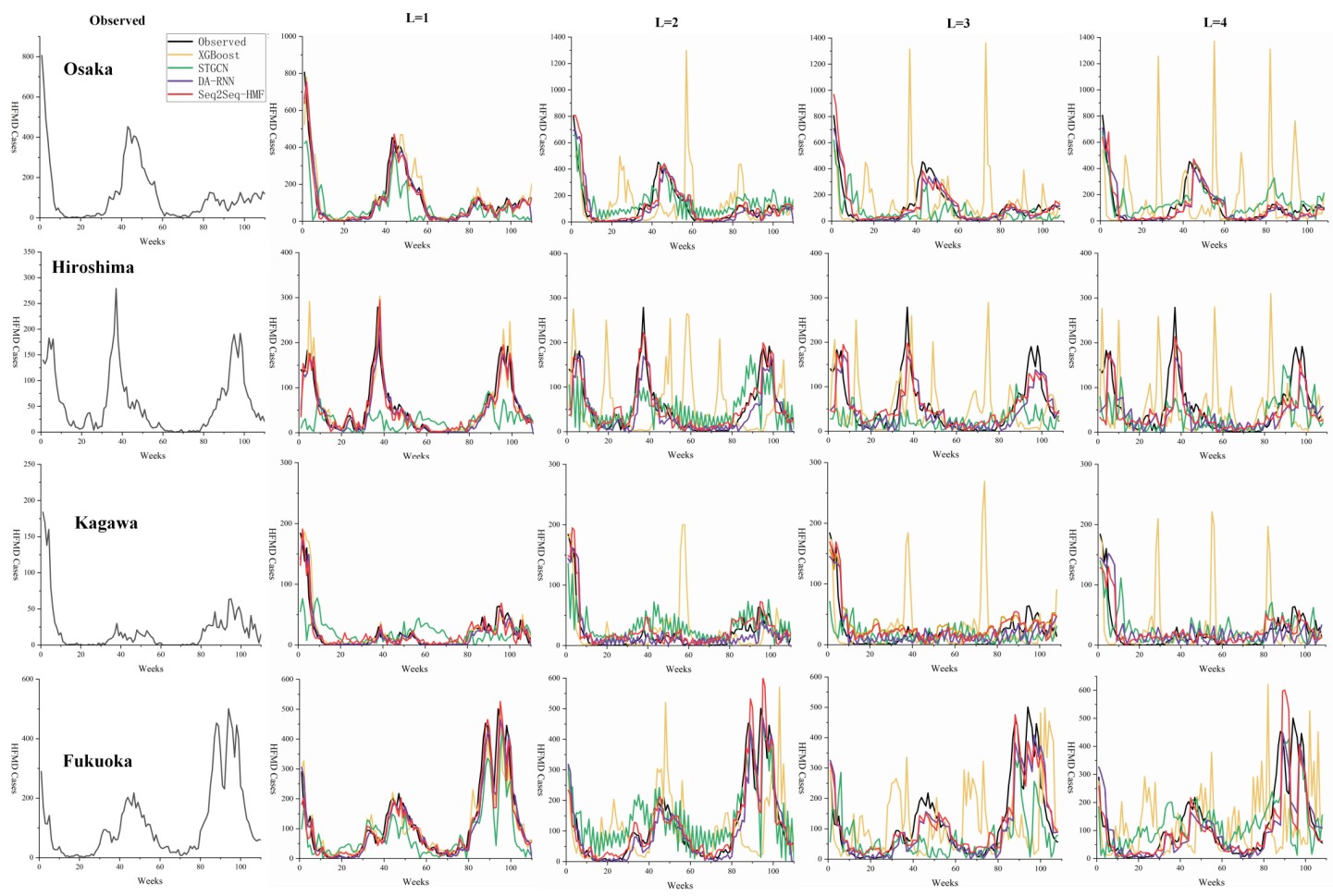

**Fig 7. Line chart of predicted values(part B).** Comparison of observed and predicted values for each representative administrative area.

and RMSE by 8.7%, while $R^2$ increased by 4.7%. Notably, at $L = 4$, $R^2$ saw the most significant increase, rising to 6.6%. The results showed that STPE cell improved the prediction performance of the model for HFMD. In EXP-3, the FECAM module was added to the basic Seq2Seq model. All metrics are also superior to those in EXP-1, particularly at $L = 3$ and $L = 4$, where the $R^2$ value increases by 6% and 7.9%, respectively, indicating that the FECAM module enhances the model's long-term prediction capability. In EXP-4, FECAM was added on top of the setup from EXP-2, leading to further improvements in the results compared to EXP-2. The $R^2$ value increased by 1.4%, 1.7%, 2.7%, and 2.3% respectively, demonstrating that FECAM enhances the accuracy of HFMD prediction and brings the model's predicted values closer to the actual ones. To visually demonstrate the impact of each submodule, a radar chart is employed to compare the evaluation metrics across different improved modules. Fig 9 indicate that Seq2Seq-HMF outperforms all other improved modules in terms of all indicators.

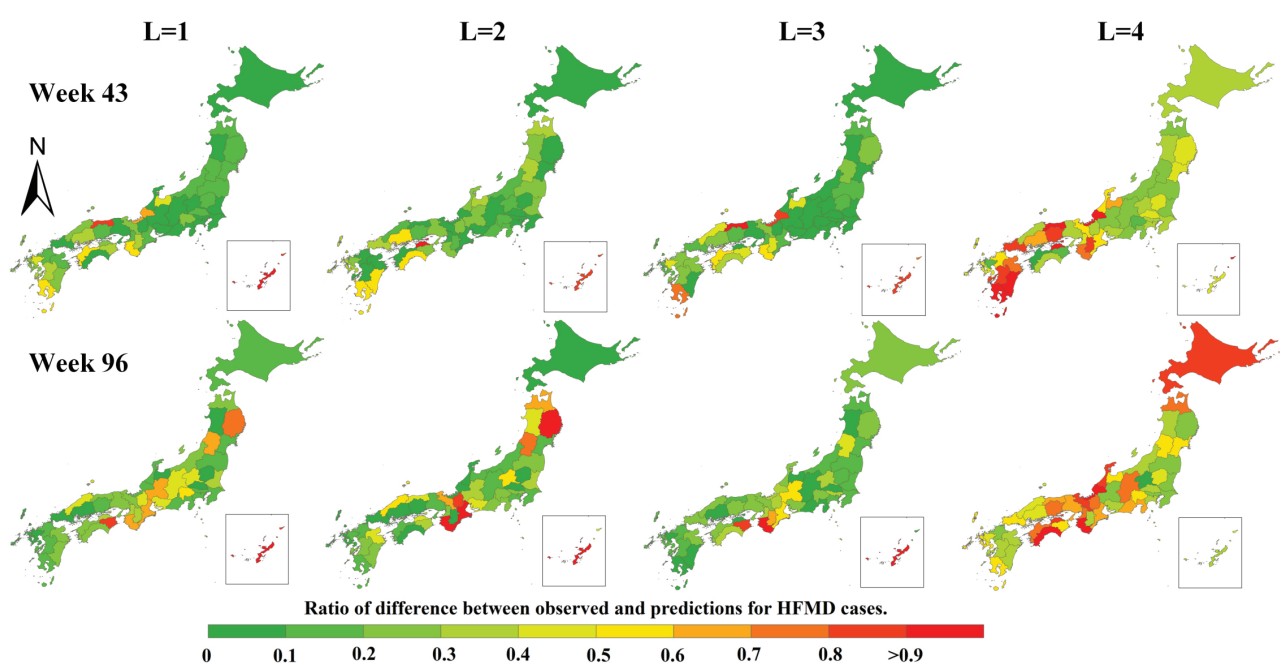

**Fig 8. Error Ratio**: A geospatial map comparing the observed values with the predictions from the Seq2Seq-HMF during the HFMD epidemic peaks.

**Table 4. Performance sensitivity to spatial weight matrix.**

| Spatial weight matix | MAE | | | | RMSE | | | | R² | | | |
|---|---|---|---|---|---|---|---|---|---|---|---|---|
| | L=1 | L=2 | L=3 | L=4 | L=1 | L=2 | L=3 | L=4 | L=1 | L=2 | L=3 | L=4 |
| Distance-based | 15.61 | 19.76 | 25.59 | 27.08 | 34.45 | 45.52 | 48.84 | 58.16 | 0.909 | 0.842 | 0.82 | 0.74 |
| Adjacency-based | **15.42** | **18.96** | **22.23** | **24.9** | **30.35** | **39.08** | **47.38** | **56.27** | **0.93** | **0.883** | **0.831** | **0.76** |

**Table 5. Performance sensitivity to sampling hop.**

| hop | MAE | | | | RMSE | | | | R² | | | |
|---|---|---|---|---|---|---|---|---|---|---|---|---|
| | L=1 | L=2 | L=3 | L=4 | L=1 | L=2 | L=3 | L=4 | L=1 | L=2 | L=3 | L=4 |
| hop=1 | 15.97 | 19.51 | 23.30 | 26.54 | 30.66 | 40.88 | **46.66** | 60.64 | 0.928 | 0.872 | **0.836** | 0.749 |
| hop=2 | **15.42** | **18.96** | **22.23** | **24.9** | **30.35** | **39.08** | 47.38 | **56.27** | **0.93** | **0.883** | 0.831 | **0.76** |
| hop=3 | 16.8 | 19.93 | 23.75 | 28.11 | 30.78 | 41.30 | 47.56 | 59.83 | 0.927 | 0.87 | 0.83 | 0.731 |
| hop=4 | 15.73 | 23.01 | 23.23 | 29.11 | 30.65 | 43.27 | 47.27 | 60.04 | 0.928 | 0.857 | 0.832 | 0.729 |
| hop=5 | 16.81 | 19.06 | 22.98 | 29.43 | 31.66 | 40.74 | 47.20 | 60.75 | 0.923 | 0.873 | 0.833 | 0.723 |

**Table 6. Performance sensitivity to regularization coefficient.**

| $\lambda$ | Mean Loss | | MAE | | RMSE | | R² | |
|---|---|---|---|---|---|---|---|---|
| | L=1 | L=4 | L=1 | L=4 | L=1 | L=4 | L=1 | L=4 |
| 0 | 0.0374 | 0.3044 | 19.44 | 28.46 | 35.42 | 60.33 | 0.904 | 0.727 |
| $1 \times 10^{-4}$ | 0.0357 | 0.3364 | 19.58 | 32.59 | 31.78 | 64.29 | 0.923 | 0.689 |
| $1.5 \times 10^{-3}$ | 0.0265 | 0.0866 | **15.42** | **24.9** | **30.35** | **56.27** | **0.93** | **0.76** |
| $3 \times 10^{-2}$ | 0.5965 | 1.996 | 48.02 | 74.45 | 118.54 | 118.34 | 0.06 | -0.54 |
| $1 \times 10^{-1}$ | 0.9564 | 2.541 | 60.18 | 74.06 | 114.29 | 118.21 | 0.002 | 0.0002 |

**Table 7. Experimental design to verify the effects of STPE cell and FECAM.**

| Experiment(EXP) | Seq2Seq | STPE cell | FECAM |
|---|---|---|---|
| EXP-1 | ✓ | × | × |
| EXP-2 | ✓ | ✓ | × |
| EXP-3 | ✓ | × | ✓ |
| EXP-4 | ✓ | ✓ | ✓ |

**Table 8. Effect of Seq2Seq-HMF component.**

| Experiment(EXP) | MAE | | | | RMSE | | | | $R^2$ | | | |
|---|---|---|---|---|---|---|---|---|---|---|---|---|
| | L=1 | L=2 | L=3 | L=4 | L=1 | L=2 | L=3 | L=4 | L=1 | L=2 | L=3 | L=4 |
| EXP-1 | 19.7 | 27.88 | 31.1 | 33.57 | 34.57 | 45.6 | 54.84 | 63.39 | 0.909 | 0.843 | 0.773 | 0.697 |
| EXP-2 | 18.65 | 19.37 | 22.3 | 26.00 | 32.96 | 41.51 | 50.08 | 58.24 | 0.917 | 0.868 | 0.809 | 0.743 |
| EXP-3 | 16.30 | 20.20 | 22.71 | 25.45 | 30.38 | 41.23 | 48.68 | 57.11 | 0.92 | 0.87 | 0.819 | 0.752 |
| EXP-4 | **15.42** | **18.96** | **22.23** | **24.90** | **30.35** | **39.08** | **47.38** | **56.27** | **0.93** | **0.883** | **0.831** | **0.76** |

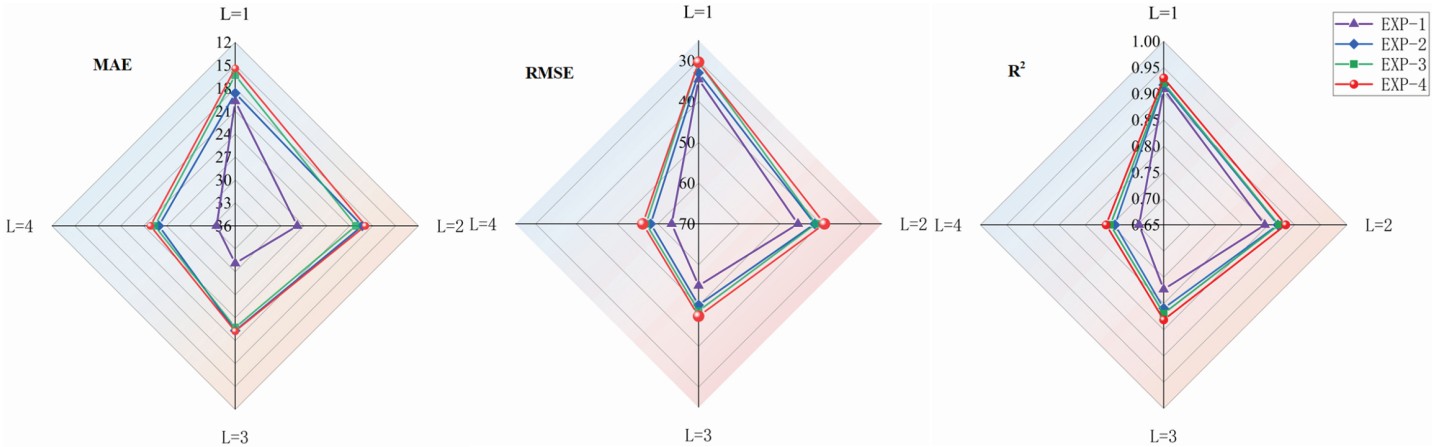

**Fig 9. Radar chart illustrating the performance metrics of the Effect of Seq2Seq-HMF component.**

## Conclusion

This study proposed a prediction model of HFMD, Seq2Seq-HMF, which performs a multi-region multi-step prediction task. The model consists of an encoder based on STPE cell and a decoder that incorporates FECAM. In the experimental section, the performance of eight HFMD prediction models was evaluated in multi-region and multi-step prediction tasks, using Japan's 47 prefectures and a Chinese city as a case study. Among these models, the Seq2Seq-HMF model demonstrated higher accuracy in predicting the number of HFMD cases for the upcoming weeks and exhibited greater precision and stability in both short- and long-term predictions.

This model offers a novel approach for HFMD prediction, aiding public health departments in accurately forecasting cases. This facilitates timely preventive measures, rational allocation of medical resources, and minimizes the impact on infants and young children. Furthermore, understanding the factors driving the model's predictions is crucial for epidemiological insights and public health decision-making. Future research could not only extend the model's application to other regions and incorporate additional social factors such as population size, mobility, and vaccine coverage rates, but also focus on developing or integrating

interpretability techniques to shed light on the spatio-temporal dependencies and key drivers identified. This would enhance both the predictive accuracy and the practical utility of the model.

## Author contributions

**Conceptualization:** Bingbing Lei.

**Data curation:** xuanjun zhu.

**Funding acquisition:** Bingbing Lei, Tao Zhou.

**Methodology:** xuanjun zhu.

**Resources:** Bingbing Lei, Tao Zhou.

**Validation:** Tao Zhou.

**Visualization:** Yuxi Zhang.

**Writing – original draft:** xuanjun zhu.

**Writing – review & editing:** Tao Zhou, Yuxi Zhang.

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
