## [Decision Letter · Decision Letter 0]

4 Apr 2025

PONE-D-25-08806Harnessing Hybrid Perception on Multi-scale Features for Hand-Foot-Mouth Disease Multi-region Prediction Based on Seq2SeqPLOS ONE

Dear Dr. zhu,

Thank you for submitting your manuscript to PLOS ONE. After careful consideration, we feel that it has merit but does not fully meet PLOS ONE’s publication criteria as it currently stands. Therefore, we invite you to submit a revised version of the manuscript that addresses the points raised during the review process.

We look forward to receiving your revised manuscript.

Kind regards,

Guangyin Jin

Academic Editor

PLOS ONE

Journal Requirements:

This work is funded by:

Bingbing Lei, Scientific research project of Ningxia Education Department(NYG2024084);

Tao Zhou, National Natural Science Foundation of China(62062003).

4. We note you have included a table to which you do not refer in the text of your manuscript. Please ensure that you refer to Table 1 in your text; if accepted, production will need this reference to link the reader to the Table.

Reviewers' comments:

Reviewer's Responses to Questions

**Comments to the Author**

1. Is the manuscript technically sound, and do the data support the conclusions?

Reviewer #1: Partly

Reviewer #2: Yes

Reviewer #3: Partly

2. Has the statistical analysis been performed appropriately and rigorously? 

Reviewer #1: Yes

Reviewer #2: Yes

Reviewer #3: Yes

3. Have the authors made all data underlying the findings in their manuscript fully available?

Reviewer #1: Yes

Reviewer #2: No

Reviewer #3: No

4. Is the manuscript presented in an intelligible fashion and written in standard English?

Reviewer #1: Yes

Reviewer #2: Yes

Reviewer #3: Yes

5. Review Comments to the Author

Reviewer #1: 1、The author lacks a clear statement of the research motivation. In the contribution section, the author directly lists the frequency-enhanced channel attention mechanism, but does not explain why this module was considered beforehand.

2、The author only discusses time series prediction models in the context of the prediction task for hand, foot, and mouth disease in the related work section. However, we know that time series prediction models are widely used and have a long research history. The author lacks an in-depth review of the relevant work.

3、The section on model structure lacks a clear and easy-to-understand diagram of the model architecture.

4、The experimental results section lacks a detailed description of the dataset and the data preprocessing methods.

5、In the experimental results section, the author does not provide an introduction to the baseline models, instead placing some of this information in the introduction section, which seems inappropriate. Additionally, all the baseline models listed by the author are deep learning models, lacking comparisons with classical hand, foot, and mouth disease prediction models and traditional machine learning models. It is recommended to include these comparisons.

6、The experimental results section only analyzes the individual experiments and lacks an overall summary of the experimental results.

7、The model should be validated on multiple datasets.

Reviewer #2: Dear Editors and Authors,

I have reviewed the manuscript and hereby put forward some suggestions for revision.

Overall, the author manages to clearly convey their work, especially the designed DL framework. However, there are still aspects that need improvement. Specifically, the accuracy of vocabulary and grammar usage requires attention. Regarding the dataset, it is rather simplistic. For instance, the data from Chengdu mentioned in Reference 21 could be incorporated for experiments and comparisons. This would enhance the comprehensiveness and validity of the research. Finally, the author is advised to explore and incorporate more SOTA DL methods. This would not only bring novelty to the study but also potentially lead to more accurate and efficient results in predicting HFMD cases.

1.Please provide the links to the data and code.

2.The abstract contains some errors and inappropriate places, please revise carefully, such as:

a.there are some grammar errors in the abstract, please check carefully.(also in the main body of this manuscript)

b.some descriptions in the abstract are unclear. For example: ‘significant information pertinent to the analysis.’

c.why is the title of the paper included in the abstract?

3.Please correct the use of proprietary terms and abbreviations, such as SVR.

4.Lines 34-42 of the main text are logically disorganized, with unclear sentence structure and poor consistency in word usage.

5.Why not attempt to test and compare the data of Chengdu city on this dataset, as mentioned in reference 21?

6.After introducing the three main contributions of the Seq2Seq-HMF model, the results of the comparative experiments and ablation experiments are directly mentioned, lacking a brief explanation of the experimental setup and objectives. It would be beneficial to supplement that these experiments were conducted to verify the effectiveness of the model and the functions of each module, enabling readers to better understand the significance of the experimental results.

7.Table 1 has not been explained in the main text. If it is necessary to list the symbols used in the text, please choose an appropriate way.

8.Since your method does not involve GNNs, I believe that there is no need for you to provide a detailed description of GNNs in the main text. A simple description is sufficient, just like the descriptions of other methods in the above text. If you want to facilitate the readers' understanding, you can choose to place the relevant content in the sup.

9.Why are the "Evaluation metrics" placed in the "Experimental results" chapter?

10.In line 295, you mentioned "a state vector depth of D = 128". Does this refer to the hidden layer parameter of the LSTM or some other parameter? Please provide a detailed explanation. Since the number of layers in a recurrent neural network generally should not be too large, an excessively large number of layers can lead to problems with gradient propagation.

11.Figure 5 looks rather chaotic. The information it contains has not been reasonably compressed but is merely presented in an intuitive manner. Please highlight the information you intend to convey. For multiple variables, you can use evaluation metrics for comparison, or display them separately and emphasize their excellent performance.

12.In Figures 6 and 7, all the lines are mixed together, making it impossible to observe useful information. Additionally, in these figures, each color represents the prediction results of a fixed model. Why not just have one legend? It seems that your figures were drawn using Python. Please integrate all the legends. In the Python code, only one plot should have the ‘plt.legend’ set.

13.In Figure 8, you might consider placing different evaluation metrics in separate graphs. Also, the form of "R2" is not reasonable. You can manually modify the text description in the graph and express R square correctly using a mathematical expression.

Reviewer #3: The manuscript PONE-D-25-08806 proposes a Seq2Seq-based deep learning model for predicting the spatial dynamics of hand, foot, and mouth disease (HFMD). While the model architecture appears innovative, the following points require clarification and improvement:

1. The title and introduction emphasize multi-region HFMD prediction as a key contribution. However, Figures 6 and 7 focus on single-region case predictions. How does simultaneous multi-region prediction fundamentally differ from separate single-region modeling? How is the claimed spatiotemporal dependency explicitly demonstrated in the current experiments? Visualizing spatial interaction patterns (e.g., using ArcGIS) to compare multi-region and single-region predictions would strengthen this claim.

2. Given that multi-region joint prediction could theoretically be achieved through conventional multi-task learning, please address: What specific advantages does the proposed complex architecture offer over simpler multi-task learning frameworks? Is the reported performance improvement statistically significant compared to baseline multi-task approaches?

3. While the model emphasizes generalization and accuracy, interpretability is critical for epidemiological applications. Please discuss: How does Seq2Seq-HMF provide insights into HFMD transmission drivers? Are there mechanisms to trace how spatiotemporal dependencies influence predictions?

4. Were input features standardized/normalized to mitigate bias? How were missing values or outliers handled? Could these methods inadvertently suppress long-term trends in the data?

5. HFMD exhibits seasonal patterns. Was periodicity explicitly modeled (e.g., via month/season encoding)? How sensitive are the results to the spatial weight matrix (e.g., adjacency-based vs. distance/population-mobility-based)? Have alternative spatial weighting schemes been tested for robustness?

6. The design of STPE encoder cells and FECAM decoder modules is presented as novel. What existing methodologies inspired these components? Are there foundational references that support the architectural choices? Citations should be provided.

7. All figures require resolution enhancement to ensure readability of labels, legends, and annotations.

6. PLOS authors have the option to publish the peer review history of their article (what does this mean?). If published, this will include your full peer review and any attached files.

Reviewer #1: **Yes: **Chao Duan

Reviewer #2: No

Reviewer #3: No

---

## [Author Response · Author response to Decision Letter 1]

14 May 2025

Response to the Comments from Reviewers

(Manuscript Number: PONE-D-25-08806)

Dear editors and anonymous reviewers:

We sincerely appreciate your thorough review of our manuscript. Your feedback and suggestions have been instrumental in improving the quality of this manuscript. After careful consideration of the suggestions provided by both the editor and reviewers, we have meticulously revised the manuscript.

Response to Journal Requirements:

Response: Our manuscript was compiled using the provided PLOS ONE LaTeX template. We managed references via BibTeX using the ‘plos2015’ style, incorporating the .bbl output into the .tex file. Figures were formatted according to PLOS ONE’s NAAS recommendations. Tables were constructed using the LaTeX template’s built-in structures. We confirm that these procedures have been followed to meet the journal’s style standards. We confirm that the manuscript adheres to these requirements as described above. Should any unexpected issues be found, please do not hesitate to let us know, and we will be happy to revise accordingly.

2. Please note that PLOS ONE has specific guidelines on code sharing for submissions in which author-generated code underpins the findings in the manuscript. In these cases, we expect all author-generated code to be made available without restrictions upon publication of the work.

Response: The code and data used in this paper have been uploaded to https://github.com/zxj550702/hfmd_predicate.

This work is funded by:

Bingbing Lei, Scientific research project of Ningxia Education Department(NYG2024084);

Tao Zhou, National Natural Science Foundation of China(62062003).

Response:The Amended Role of Funder Statement:

Bingbing Lei, Scientific research project of Ningxia Education Department (NYG2024084): This funder supported the study and contributed to the study design through providing ideas for the research framework.

Tao Zhou, National Natural Science Foundation of China (62062003): This funder supported the study and contributed to the data analysis and figure preparation by providing guidance.

The above statement has been added in the coverletter

4.We note you have included a table to which you do not refer in the text of your manuscript. Please ensure that you refer to Table 1 in your text; if accepted, production will need this reference to link the reader to the Table.

Response: Thank you for the correction. In the revised manuscript, we refer to the Table 1 in section Graph convolution neural network. Because this is where a lot of symbols are mentioned.

Response to Reviewer #1:

Concern 1-1: The author lacks a clear statement of the research motivation. In the contribution section, the author directly lists the frequency-enhanced channel attention mechanism, but does not explain why this module was considered beforehand.

Response: Based on your suggestions, we have revised the description of the research motivation in both the abstract and the fourth paragraph of the introduction, making it more concise and precise. Additionally, to supplement the mention of “Furthermore, existing models overlook the utilization of frequency-domain features, which harbor significant information pertinent to the analysis.” in the abstract, we have elaborated on the rationale for introducing this mechanism in the fifth paragraph of the introduction.

Abstract

Accurate prediction of Hand, Foot, and Mouth Disease (HFMD) is crucial for effective epidemic prevention and control. Existing prediction models often overlook the cross-regional transmission dynamics of HFMD, limiting their applicability to single regions. Furthermore, their ability to perceive spatio-temporal features holistically remains limited, hindering the precise modeling of epidemic trends. To address these limitations, a novel HFMD prediction model named Seq2Seq-HMF is proposed, which is based on the Sequence-to-Sequence(Seq2Seq) framework. This model leverages hybrid perception of multi-scale features...

Introduction

Paragraph 6

Nevertheless, the spread of HFMD is a dynamic spatial process, characterized by randomness, uncertainty, and intricate spatio-temporal fluctuations. Traditional prediction methods often struggle to accurately capture the characteristics of multi-regional HFMD epidemic transmission simultaneously. And these methods often overlook the extraction of frequency domain features. Frequency domain features can unveil the periodicity, seasonality, and relative intensity of various frequency components, which are challenging to directly observe in the time domain. These features offer crucial information that enhances the prediction accuracy of HFMD models. Therefore, it is crucial to develop a methodology that effectively captures nonlinear associations and spatio-temporal dependencies to improve prediction accuracy. Specifically, the core challenge is developing a model that effectively integrates multiple factors, enabling it to capture spatio-temporal dependencies, perform spatio-temporal inference, and account for key influencing variables concurrently. Moreover, simultaneous prediction of epidemic trends across multiple regions, along with multi-step prediction(e.g. , short-, medium-, and long-term), would better support the dynamic allocation of prevention and control resources and the development of emergency plans.

Concern 1-2: The author only discusses time series prediction models in the context of the prediction task for hand, foot, and mouth disease in the related work section. However, we know that time series prediction models are widely used and have a long research history. The author lacks an in-depth review of the relevant work.

Response: Thank you for your detailed addition to the completeness of the related work. In the revised manuscript, we have incorporated a more comprehensive analysis and discussion of the existing literature pertinent to time series prediction models.

Concern 1-3: The section on model structure lacks a clear and easy-to-understand diagram of the model architecture.

Response: Thanks for your valuable suggestion. In response to your suggestion, we modified the model description in the Model Architecture section and provided additional details in the notes for Fig. 1 and Fig. 2.

Model architecture

Fig. 1 The overall structure of Seq2Seq-HMF. It comprises an encoder and a decoder. The encoder performs multi-scale hibrid perception on input time series data from multiple regions to extract features, which the decoder then processes to generate predictive time series data for those regions.

Fig. 2 Structure of STPE Cell. It is contains TGCN (A) and BiLSTM (B). Both modules receive the output of the previous STPE Cell in addition to the data features.

Concern 1-4: The experimental results section lacks a detailed description of the dataset and the data preprocessing methods.

Response: Thanks for your valuable feedback. In response to your suggestion, we have relocated the data collection section from the Methods section to the Experiments section. Additionally, we have enriched this section with description of the data preprocessing methods employed.

Concern 1-5: In the experimental results section, the author does not provide an introduction to the baseline models, instead placing some of this information in the introduction section, which seems inappropriate. Additionally, all the baseline models listed by the author are deep learning models, lacking comparisons with classical hand, foot, and mouth disease prediction models and traditional machine learning models. It is recommended to include these comparisons.

Response: Thanks for your constructive comments. According to your comments, we added the description of the baseline model in the comparison test section, and added the Random Forest Regression(RFR) and Extreme Gradient Boosting(XGBoost) models in the machine learning model into the experiment as comparison experiments. Based on the experimental results, Table 3, Figure 4, Figure 5 and Figure 6 were modified.

Concern 1-6: The experimental results section only analyzes the individual experiments and lacks an overall summary of the experimental results.

Response: Thanks for your valuable feedback. In the revised manuscript, the conclusion section has been extended to improve clarity by providing a more detailed summary of the experimental results.

Conclusion

Paragraph 1

This study proposed a prediction model of HFMD, Seq2Seq-HMF, which performs a multi-region multi-step prediction task. The model consists of an encoder based on STPE cell and a decoder that incorporates FECAM. In the experimental section, the performance of eight HFMD prediction models was evaluated in multi-region and multi-step prediction tasks, using Japan’s 47 prefectures and a Chinese city as a case study. Among these models, the Seq2Seq-HMF model demonstrated higher accuracy in predicting the number of HFMD cases for the upcoming weeks and exhibited greater precision and stability in both short- and long-term predictions.

Concern 1-7: The model should be validated on multiple datasets.

Response: We appreciate the reviewer’s suggestion regarding the validation of our model on multiple datasets. We have addressed this concern by evaluating our proposed Seq2Seq-HMF model using diverse datasets, as detailed below:

Multi-Region Dataset (Japan): The primary dataset used for evaluating the multi-region, multi-step prediction capability of Seq2Seq-HMF consists of weekly Hand, Foot, and Mouth Disease (HFMD) case counts from December 2013 to December 2023 for all 47 prefectures of Japan. This data, obtained from the National Institute for Infectious Diseases (NIID), was complemented with corresponding weekly meteorological data (temperature, relative humidity, atmospheric pressure, wind speed, and rainfall) sourced from the Japan Meteorological Agency (JMA). This comprehensive multi-source dataset allows us to assess the model’s ability to capture spatiotemporal dependencies across different geographical areas.

Single-Region Dataset (China): To provide a different perspective on the model’s performance and its generalizability to different data granularities and geographical contexts, we additionally utilized a single-region dataset. This dataset comprises daily HFMD case counts and meteorological variables for a specific city in southern China, spanning from January 2014 to December 2019. This publicly available dataset, released by the China National Center for Disease Control and Prevention, allows us to evaluate the model’s performance on daily data and in a different epidemiological setting.

By validating Seq2Seq-HMF on both a multi-regional, weekly dataset from Japan and a single-regional, daily dataset from China, we demonstrate the model’s effectiveness across different spatial scopes, temporal resolutions, and geographical locations.

Response to Reviewer #2:

Overall, the author manages to clearly convey their work, especially the designed DL framework. However, there are still aspects that need improvement. Specifically, the accuracy of vocabulary and grammar usage requires attention. Regarding the dataset, it is rather simplistic. For instance, the data from Chengdu mentioned in Reference 21 could be incorporated for experiments and comparisons. This would enhance the comprehensiveness and validity of the research. Finally, the author is advised to explore and incorporate more SOTA DL methods. This would not only bring novelty to the study but also potentially lead to more accurate and efficient results in predicting HFMD cases.

Response: Thank you for your positive feedback on our manuscript. We have provided point-by-point responses to your concerns, and we hope they address any remaining issues. In addition, we have thoroughly revisited certain parts of the paper in light of your comments and made enrichments and revisions accordingly.

Concern 2-1: Please provide the links to the data and code.

Response: The code and data used in this paper have been uploaded to https://github.com/zxj550702/hfmd_predicate.

Concern 2-2: The abstract contains some errors and inappropriate places, please revise carefully, such as:

a.there are some grammar errors in the abstract, please check carefully.(also in the main body of this manuscript)

b.some descriptions in the abstract are unclear. For example: ‘significant information pertinent to the analysis.’

c.why is the title of the paper included in the abstract?

Response: Thank you for pointing out these specific areas for improvement in the abstract. We have carefully reviewed your comments regarding grammatical errors, unclear descriptions, and the inclusion of the paper title. We have revised the abstract accordingly to correct these issues, improve clarity, and ensure it is concise and informative. We have also checked the main manuscript body for similar grammatical errors and made necessary corrections throughout. All changes, including these revisions to the abstract, are clearly marked in the file ‘Revised Manuscript with Track Changes’ for your review.

Concern 2-3: Please correct the use of proprietary terms and abbreviations, such as SVR.

Response: Thank you for the details. In the revised manuscript, “Support Vector Regression(SVR)” was explained in introduction section. And we have proofread more of the use of proprietary terms and abbreviations to make them accurate.

Concern 2-4: Lines 34-42 of the main text are logically disorganized, with unclear sentence structure and poor consistency in word usage.

Response: Thank you for the details. In the revised manuscript, Lines 34-42 of the main text have been revised to address issues of confused logic, unclear sentence structure, and poor word consistency.

Concern 2-5: Why not attempt to test and compare the data of Chengdu city on this dataset, as mentioned in reference 21?

Response: We appreciate the suggestion to test our model on the Chengdu city data mentioned in reference 21. Regarding this specific dataset, we found that it is not publicly available, which limits its direct use for validation in this study. Through our efforts to obtain diverse data for comprehensive validation, we acquired the following datasets, as detailed below:

This dataset comprises daily HFMD case counts and meteorological variables for a specific city in southern China, spanning from January 2014 to December 2019. This publicly available dataset, released by the China National Center for Disease Control and Prevention, allows us to evaluate the model’s performance on daily data and in a different epidemiological setting.

Concern 2-6: After introducing the three main contributions of the Seq2Seq-HMF model, the results of the comparative experiments and ablation experiments are directly mentioned, lacking a brief explanation of the experimental setup and objectives. It would be beneficial to supplement that these experiments were conducted to verify the effectiveness of the model and the functions of each module, enabling readers to better understand the significance of the experimental results.

Response: Thank you for your constructive comments. We appreciate the suggestion to clarify the experimental setup and objectives. In the revised manuscript, these details have been explained in the ‘Experimental Setup’ section. This section now clearly outlines the purpose of the comparative and ablation experiments, which were designed to verify the overall effectiveness of the Seq2Seq-HMF model and the specific functions of its individual modules, thereby enhancing the reader’s understanding of the subsequent results.

Experimental setup

Paragraph 2

Comparative e

---

## [Decision Letter · Decision Letter 1]

27 May 2025

Harnessing Hybrid Perception on Multi-scale Features for Hand-Foot-Mouth Disease Multi-region Prediction Based on Seq2Seq

PONE-D-25-08806R1

Dear Dr. zhu,

We’re pleased to inform you that your manuscript has been judged scientifically suitable for publication and will be formally accepted for publication once it meets all outstanding technical requirements.

Kind regards,

Guangyin Jin

Academic Editor

PLOS ONE

Additional Editor Comments (optional):

Reviewers' comments:

Reviewer's Responses to Questions

**Comments to the Author**

1. If the authors have adequately addressed your comments raised in a previous round of review and you feel that this manuscript is now acceptable for publication, you may indicate that here to bypass the “Comments to the Author” section, enter your conflict of interest statement in the “Confidential to Editor” section, and submit your "Accept" recommendation.

Reviewer #1: All comments have been addressed

Reviewer #3: All comments have been addressed

2. Is the manuscript technically sound, and do the data support the conclusions?

Reviewer #1: Yes

Reviewer #3: Yes

3. Has the statistical analysis been performed appropriately and rigorously? 

Reviewer #1: Yes

Reviewer #3: Yes

4. Have the authors made all data underlying the findings in their manuscript fully available?

Reviewer #1: Yes

Reviewer #3: Yes

5. Is the manuscript presented in an intelligible fashion and written in standard English?

Reviewer #1: Yes

Reviewer #3: Yes

6. Review Comments to the Author

Reviewer #1: The author has effectively addressed the relevant questions, enriched the experimental section, and revised the results section, making the article more persuasive.

Reviewer #3: After thoroughly evaluating the revised manuscript and the authors' comprehensive responses, I am pleased to recommend that this paper be accepted for publication.

7. PLOS authors have the option to publish the peer review history of their article (what does this mean?). If published, this will include your full peer review and any attached files.

Reviewer #1: **Yes: **Chao Duan

Reviewer #3: No

---

## [Editor Report · Acceptance letter]

PONE-D-25-08806R1

PLOS ONE

Dear Dr. zhu,

I'm pleased to inform you that your manuscript has been deemed suitable for publication in PLOS ONE. Congratulations! Your manuscript is now being handed over to our production team.

Kind regards,

on behalf of

Dr. Guangyin Jin

Academic Editor

PLOS ONE